# Ovarian Cancer and Pregnancy—A Current Problem in Perinatal Medicine: A Comprehensive Review

**DOI:** 10.3390/cancers12123795

**Published:** 2020-12-16

**Authors:** Dominik Franciszek Dłuski, Radzisław Mierzyński, Elżbieta Poniedziałek-Czajkowska, Bożena Leszczyńska-Gorzelak

**Affiliations:** Chair and Department of Obstetrics and Perinatology, Medical University of Lublin, 20-954 Lublin, Lubelskie Region, Poland; radek@bg.umlub.pl (R.M.); elzbieta.poniedzialek@umlub.pl (E.P.-C.); b.leszczynska@umlub.pl (B.L.-G.)

**Keywords:** adnexal masses, management in pregnancy, ovarian cancer, ovarian malignancy, ovarian tumor, pregnancy

## Abstract

**Simple Summary:**

Nowadays, the number of malignancies diagnosed during pregnancy is increasing. Despite the fact that diagnosis is occurring on a global scale, their number is still too limited to prepare proper standards of treatment. These problems appear specifically in the least developed countries. The aim of our review is to bring ovarian cancer (OC) as a complication of pregnancy to the attention of doctors and other medical professionals who have to cope with these rare cases. We noted that a variety of malignancies can be included under the heading “ovarian cancer”, and we describe obstetric and patient outcomes, which depend on the histopathology of the tumor. We focus on the current recommendations for diagnostics and treatment, and present future possibilities for the management of OC.

**Abstract:**

The frequency of concomitant adnexal tumors in pregnancy is reported to be at 0.15–5.7%, while ovarian cancer complicates 1 in 15,000 to 1 in 32,000 pregnancies, being the second most common gynecologic cancer diagnosed during pregnancy. The aim of this review is to discuss the problem of ovarian cancer complicating pregnancy and the current recommendations for diagnostics and treatment, with an emphasis on the risk to the fetus. A detailed analysis of the literature found in the PubMed and MEDLINE databases using the keywords “ovarian cancer”, “ovarian malignancy”, “adnexal masses”, “ovarian tumor” and “pregnancy” was performed. There were no studies on a large series of pregnant women treated for ovarian malignancies and the management has not been well established. The diagnostics and therapeutic procedures need to be individualized with respect to the histopathology of the tumor, its progression, the gestational age at the time of diagnosis and the mother’s decisions regarding pregnancy preservation. The multidisciplinary cooperation of specialists in perinatal medicine, gynecological oncology, chemotherapy, neonatology and psychology seems crucial in order to obtain the best possible maternal and neonatal outcomes.

## 1. Introduction

Because of the delay due to planning pregnancy at a later reproductive age and the fact that the frequency of the incidence of numerous neoplasms increases in the fourth decade of life, the number of pregnant women affected by cancer is rising [1,2]. This challenging problem complicates 1 in 1000 pregnancies and is becoming more common nowadays [3,4,5]. The frequency of adnexal tumors in pregnancy is reported to be 0.15–5.7%, with most being benign [6]. The transformation of benign ovarian tumors such as dermoid cysts into ovarian cancer (OC) is very rare and just four cases have been reported in the literature devoted to this subject [7,8,9,10]. Ovarian cancer is the second most common gynecologic cancer diagnosed during pregnancy, complicating 1 in 15,000 to 1 in 32,000 pregnancies [6,11,12]. OC represents 3–6% of neoplasms and 49–75% of ovarian malignancies in pregnancy [1,13,14]. According to the statistics, ovarian cancer takes the fifth position among the most common malignancies affecting pregnant women, following breast cancer, thyroid cancer, cervical cancer and Hodgkin lymphoma [15]. However, one of the reviews reported that in the Asian population, ovarian cancer takes the sixth position [16].

The morbidity and mortality rates of ovarian cancer have fallen in recent years due to an increased reliance on oral contraceptives, but it still takes seventh place among the most common malignancies in women worldwide and third place for the most common gynecological cancer [17]. In 2018, 240,000 new cases were diagnosed, resulting in a rate of incidence of 10–15 per 100,000, with the highest mortality rate among female cancers. It is the fifth leading cause of death in women worldwide [18]. The 5-year survival rates depend on the advancement of cancer and vary from 92% in Stage I to 30% in Stage IV. The term “ovarian cancer” refers to a neoplasm that can originate in the ovary, fallopian tube, peritoneum and other tissues of the pelvis, where the site cannot be assigned to one of the aforementioned ones [19]. A variety of malignancies such as epithelial ovarian cancers (EOCs), including borderline tumors, germ cell tumors and sex cord tumors, might also be included under the umbrella of “ovarian cancer”.

EOCs encompass the vast majority of ovarian cancers of non-pregnant women (approximately 90%), but it is responsible for fewer cases of OC during pregnancy [20] (Table 1). Much more common are borderline, germ cell and sex cord tumors, which appear to be chiefly responsible for the increased prevalence during reproductive age [21]. Borderline tumors are neoplasms of low malignant potential, but in pregnant women, they present a higher incidence of more aggressive histologic features, such as microinvasion [22]. The two most common germ cell tumors in pregnancy are dysgerminoma and yolk-sac tumor [21]. Five subtypes of epithelial ovarian cancers can be distinguished: high-grade serous (70%), low-grade serous (<5%), endometrioid (10%), clear cell (10%) and mucinous (3%) ovarian cancer. An alternative classification of EOC categorizes it into two main groups: Type I and Type II, with Type I being less common (30%) and diagnosed in earlier stages [20]. Most ovarian cancers during pregnancy present a good prognosis, due to early diagnosis in the first stage of the disease [23]. The 5-year survival rate for ovarian tumors complicating pregnancy is estimated to be 72–90% [24].

### Aim

The aim of this review was to present the problem of ovarian cancer complicating pregnancy and the current recommendations for diagnostics and treatment, with an emphasis on the risk to the fetus. This review also presents future possibilities for OC management.

## 2. Material and Methods

A literature search in the electronic databases PubMed and MEDLINE was performed. We focused on pregnancy complicated by ovarian cancer. A detailed analysis of eligible publications in the literature using MESH terms such as “ovarian cancer”, “ovarian malignancy”, “adnexal masses”, “ovarian tumor” and “pregnancy” as keywords was conducted. Only publications in English were considered. The references included in these selected publications were also taken into account with the aim of finding additional relevant articles. We analyzed the following types of articles: case reports, clinical trials, population-based studies, reviews, systematic reviews and meta-analyses, due to the rarity of the discussed problem.

## 3. Results

A computerized literature search in the electronic databases PubMed and MEDLINE was performed to identify articles on ovarian cancer and pregnancy using the keywords “ovarian cancer”, “ovarian malignancy”, “adnexal masse”, “ovarian tumor” and “pregnancy”. Only publications in English were considered. Since the discussed problem is extremely rare, the following types of articles were analyzed: case reports, clinical trials, population-based studies, reviews, systematic reviews and meta-analyses. We initially identified a total of 1319 papers. Next, the duplicates were removed and 551 articles were left for our assessment, of which 414 articles were excluded based on the titles and abstracts, leaving 137. After reading the full-text articles, 81 papers were excluded from the analysis because of incomplete case reporting or the inability to assign individual diagnostics, clinicopathological data, treatment regimens or outcomes. We also excluded articles devoted to only the technical aspects of surgical management and detailed histopathological analyses. The final results of the review of 56 articles are presented in five sections: diagnostics, management, obstetric outcomes, patient outcomes and perspectives.

### 3.1. Diagnostics

Diagnostics start with anamnesis and clinical examination, which might be much more difficult during pregnancy but can still be performed without any limitations. The presenting symptoms of ovarian cancer in pregnant women are the same as in non-pregnant women but can be overlooked as being related to ovarian cancer because they overlap and are therefore attributed as being symptoms that result from the physiological changes during pregnancy. Fatigue, anemia, nausea, vomiting, increased abdominal circumference, constipation, shortness of breath or urinary symptoms can be confounded with pregnancy, which results in delayed diagnosis [28,29]. However, during pregnancy, emergency abdominal events such as torsion of an ovarian mass, rupture of an ovarian mass or intraperitoneal hemorrhage are common [30]. In some pregnant women, a suspicious adnexal mass, a cul-de-sac mass or a nodularity can be found during an antenatal physical examination.

Although blood examinations are not limited, some of them should be considered with caution, such as those regarding tumor markers, the levels of which could be elevated during gestation. It is particularly important to take into account the fact that levels of cancer antigen 125 (CA 125) and human chorionic gonadotropin (hCG) in blood are typically increased during gestation, especially in the first trimester. Increased levels of inhibin B, lactate dehydrogenase (LDH), human epididymis protein 4 (HE4), cancer antigen 19-9 (CA 19-9), carcinoembryonic antigen (CEA) and anti-Müllerian hormone (AMH) can be useful in the diagnosis of ovarian cancer as a pregnancy complication, since their levels are expected to be normal during gestation [23,29,31]. Unexplained high levels of alpha-fetoprotein (AFP) and inhibin A during antenatal screening can be confused with indications of neural tube defects or Down’s syndrome and may also be the first signs of the presence of adnexal masses [29].

Different types of imaging tests during pregnancy may be considered, e.g., ultrasonography (USG), magnetic resonance imaging (MRI) and computed tomography (CT). USG and MRI do not include ionizing radiation and are not harmful to the embryo or fetus. Prenatal USG as a routine procedure contributes to the increased number of diagnosed asymptomatic ovarian masses [12]. The most common model used to evaluate pelvic masses in non-pregnant women is the International Ovarian Tumor Analysis (IOTA) Group Simple Rules [32] (Table 2), though the usefulness of this scoring system has not yet been recognized in pregnant women. When the ovarian mass is too big to access or the ultrasonography diagnosis is inconclusive, or there exists an increased risk of malignancy, further imaging examination is needed. The most optimal second-line imaging in pregnant women is MRI, which is highly accurate in examining the characteristics of complex or indeterminate ovarian masses, or in presurgical evaluation of the extent of disease, peritoneal dissemination and nodal metastases [33,34]. The Assessment of Different NEoplasias in the adneXa magnetic resonance (ADNEX MR) scoring system allows the detection of cancer with an overall accuracy higher than 80% [35] (Table 3). Pineapple juice can be used as a negative contrast for MRI; indeed, it is useful in the investigation of peritoneal/intra-abdominal lesions and adhesions, especially in evaluating ovarian cancer. Additionally, pineapple juice is safe for the mother and child, which is crucial for pregnant women [36]. Computed tomography (CT) is not recommended during pregnancy because it could be related to potential teratogenic effects. Despite the fact that the radiation dose absorbed by the fetus is lower than the dose needed to damage the baby, the scholastic effect of ionizing radiation may occur in intensely dividing fetal cells, which can lead to the development of a neoplasm in childhood [37,38]. 

### 3.2. Management

The current management of ovarian cancer involves surgery, chemotherapy and radiotherapy. There are no definitive guidelines in the literature regarding the management of ovarian cancer in pregnancy. Earlier publications recommended that pregnant women with ovarian cancer should be treated in the same way as non-pregnant women with immediate laparotomy, regardless of the duration of pregnancy and the condition of the mother and child [40,41,42,43,44,45]. Nowadays, the management depends on a few factors such as the duration of pregnancy, the general condition of the mother and fetus, and the mother’s desire to continue the pregnancy.

Surgery seems to be the least controversial type of oncologic management in pregnancy, as there have been no adverse effects reported for surgical treatment for non-oncological cases. Non- obstetrical surgery during pregnancy is performed in 1–4 out of 200 cases. Ovarian cancer surgery during gestation requires special attention, as for abdominal approaches. Two things seem important: optimal surgical outcomes and the safety of the mother and fetus [46]. A very careful decision must be made with regard to performing an operation in adequate time, obviously not too early (because of the risk of loss of luteal function by the ovary before the fourth month of gestation and miscarriage) and not too late (progression of malignancy, preterm labor, ovarian torsion, rupture or bleeding). An adnexal tumor found incidentally during cesarean section should be removed [47,48]. The general ovarian tumor consensus management is that surgery is needed when the adnexal mass is:Larger than 10 cm in diameter;Persists into the second trimester; orPresents solid or mixed cystic and solid, highly suspicious characteristics during ultrasound [11,47,49,50,51].

The optimal time for surgical treatment during pregnancy is early in the second trimester (between 16 and 20 weeks of gestation). There are numerous reasons for this:Organogenesis is complete, which minimizes the risk of teratogenesis induced by medications;The placenta replaces the hormonal function of the corpus luteum and resection does not affect progesterone concentration;Low risk of pregnancy loss related to second trimester surgery;By this time, almost all functional cysts will have been resolved;Spontaneous miscarriages connected with fetal abnormalities are likely to have already occurred and will not be mistakenly attributed to the surgical treatment [52].

If surgery in a pregnant woman is required before 14 weeks of gestation, when the placenta becomes capable of producing hormones in sufficient levels, progesterone should be administered in doses of 60–120 mg i.m./day or 300–600 mg vag. or p.o./day [53].

Laparoscopy is feasible during pregnancy but depends on the following rules:The operator has proper experience in performing laparoscopy during pregnancy;The optimal time for laparoscopy, which is 16–20 weeks of gestation;Trocars’ localization, which depends on gestational age; the first should be inserted at least 3–4 cm above the fundus of the uterus;Must be no longer than 90–120 min;Low abdominal pressure: 10 and 13 mm Hg;The preferred technique is open introduction without a Veress needle [54,55,56,57].

A comparison between laparoscopy and laparotomy in pregnant women has recently been published. It was found that laparoscopy is associated with shorter hospital stays, shorter operative times and fewer adverse effects for the fetus [58]. In addition to this, pregnant patients undergoing laparotomy for an adnexal tumor have more frequent preterm contractions than patients undergoing laparoscopic surgery [59]. We have to bear in mind that laparoscopy may cause perforation of the uterus, hypercapnia and reduced blood flow due to increased abdominal pressure and use of carbon dioxide [55,56,57]. 

Complete surgical resection of the neoplasm is preferred over aspiration and cytologic evaluation. In most cases, there will be an inappropriate incision for surgical staging, so a frozen section during operation is needed to decide about the scope of the operation and later management [53,60] (Figure 1 and Figure 2). The staging procedure during pregnancy can include appendectomy, infracolic omentectomy, pelvic peritoneal biopsies or lymph node dissection. There is a general recommendation that restaging should be planned postpartum if the pouch of Douglas and the pelvic peritoneum cannot be reliably examined during the operation in pregnant women. Based on the opinion of experts, a proper gynecological surgical assessment may be proposed at approximately 22 weeks of gestation. If a low progression to invasive cancer is diagnosed during the second or third trimester in an adnexal mass with a low malignant potential, the surgery can be postponed until postpartum. If the diagnosis of advanced epithelial ovarian cancer is made in the first part of pregnancy, termination of the pregnancy should be considered and discussed with the patient. If the pregnant woman decides on the preservation of pregnancy, a biopsy or adnexectomy with subsequent platinum-based chemotherapy should be recommended. In cases such as this, debulking (cytoreductive) surgery should be planned after delivery, since this type of surgery cannot be performed during pregnancy [53] (Figure 1 and Figure 2).

Chemotherapy (CT) is the basis for adjuvant and neoadjuvant therapy of OC but it is linked with a number of problems that are not observed in non-pregnant women, such as spontaneous abortion, congenital abnormalities, or hypotrophy [6]. The highest risk (10–20%) for congenital malformations is noted between 4 and 10 weeks of gestation, so chemotherapy is recommended from the second trimester onward, when fetal organogenesis is complete [61,62]. Beyond 35 weeks of gestation, CT is not recommended, as the three-week window between the last course of chemotherapy and delivery is needed for the recovery of both fetal and maternal bone marrow, which should decrease the possibility of bleeding, infection or anemia with avoidance drug accumulation in the fetus. This is especially important for preterm babies, who do not have the enzymes to metabolize CT adequately. Most chemotherapeutics involve relatively small molecules that might cross the placenta [63]. The administration of chemotherapy is dependent on the histological examination. In the case of borderline tumors, there are no indications for adjuvant therapy [33,53]. Chemotherapy is also not recommended in epithelial OC Stage IAG1, dysgerminoma Stage I, immature teratoma Stage IG1, or folliculoma Stages IA and IB with no adverse risk factors [54,64]. In epithelial ovarian cancer, chemotherapy consists of the administration of paclitaxel or docetaxel and carboplatin or cisplatin. Germ cell tumors and stromal cell tumors are treated with paclitaxel and carboplatin (first-line treatment) or with a bleomycin, viblastine and cisplatin (BVP) or cisplatin, etoposide, bleomycin (PEB) regimen (second-line treatment). First-line treatment with paclitaxel and carboplatin is preferred, as it has the most favorable safety prolife for the fetus and these drugs have proven effectiveness [65,66] (Table 4).

Targeted therapy, which is used in OC cases in non-pregnant women, should also be discussed; the data are limited to case reports and case series in humans [67,68]. Two main questions need to be answered: (1) How might the targeted therapies influence fetal development? (2) How do we cope with such a great number of different drugs with different pharmacological properties? Small molecules such as poly(ADP-ribose), polymerase (PARP) inhibitors, or tyrosine kinase inhibitors (TKIs) have the capacity to cross through the placenta throughout the entire pregnancy. Large molecules (for example, monoclonal antibodies, which require active transport through the placenta) can reach the fetus after 14 weeks of gestation [53]. Research on animals demonstrated their potential embryotoxicity and risk of adverse fetal outcomes [69]. Angiogenesis inhibitors such as bevacizumab, a humanized anti-vascular endothelial growth factor (VEGF) antibody, are teratogenic and induce intrauterine growth restriction (IUGR), pregnancy loss and skeletal malformation in animal models (mice and rats). As is well known, angiogenesis is crucial for the development of the fetus and the placenta, so bevacizumab and other antiangiogenic drugs are definitely contraindicated in pregnancy. There are no cases of the systemic administration of PARP inhibitors, other targeted agents or immunotherapy during pregnancy in humans with ovarian cancer or other gynecological malignancies. Due to limited evidence for the clinical use and safety of targeted therapies, it is recommended to postpone their use until after delivery [53].

Radiotherapy (RT) denotes the use of high-energy radiation from X-rays, gamma rays, neutrons, protons, and other sources to kill cancer cells and to restrict tumors. The influence of RT on pregnancy is generally recognized and might involve fetal death, fetal abnormalities and growth disturbances, and may lead to carcinogenesis during pregnancy or in childhood. These effects depend on the gestational age during exposure and radiation dose [70,71]. There is no place for radiotherapy during pregnancy for gynecologic cancers, unless embryonic or fetal death is considered unavoidable [37,72].

### 3.3. Obstetric Outcomes

All pregnant women with ovarian cancer have high-risk pregnancies and need a specially dedicated and well-equipped perinatal and oncological center. At the same time, the standard screening, diagnostics for chromosomal and structural defects or other pregnancy complications cannot be neglected. Ultrasound fetal examination with the assessment of growing intervals, amniotic fluid index (AFI), cervical length and Doppler exam of blood flows in the middle cerebral artery (MCA) and umbilical artery (UA) seems obligatory [73]. The assessment of the peak systolic velocity (PSV) in the MCA is of great clinical value in the care of high-risk pregnancies for the diagnosis and management of fetal anemia and intrauterine growth restriction (IUGR) [74]. Supplementation of folic acid and nutritional counseling is also essential to optimize maternal and fetal status. Pregnant patients with surgical treatment have better obstetric outcomes than women with CT or surgery + CT. There are fewer cases of admission to neonatal intensive care units (NICUs), IUGR, preterm premature rupture of the membranes (pPROM), preterm contractions or neonates being small for their gestational age [73]. If possible, the patient should not be delivered before 37 weeks, which can help to avoid neonatal morbidities related to prematurity. When preterm delivery is inevitable, the administration of steroids should be considered [53]. After delivery, the placenta should also be examined histopathologically for metastases; some gynecological cancers are exceptionally related to metastases in the child and placenta [75,76,77,78]. 

### 3.4. Patient Outcomes

In a descriptive cohort study, 1170 pregnant women with cancer were reported by de Haan et al. with results from over 20 years of research; 37 centers from 16 countries participated in the International Network on Cancer, Infertility and Pregnancy (INCIP). The researchers registered and analyzed data on obstetric, maternal, neonatal and oncological outcomes. In most cases (893 patients), the diagnosed cancer was in local or regional advancement. In this report, ovarian cancer was diagnosed in 7% of cases (88 patients), which corresponds to the fourth position among all types of cancer concomitant with pregnancy. The increased frequency of OC can be observed with respect to previous statistics. No women in that study died due to ovarian cancer during pregnancy [73]. Lee et al. analyzed the most common malignancies associated with pregnancy in Australia. The researchers defined them as cancers with an initial diagnosis made during pregnancy or in a period of 12 months after delivery. These data were collected in New South Wales between 1994 and 2008, and cancer was diagnosed in 1798 patients (499 during pregnancy and 1299 postpartum) being, in most cases, in local or regional advancement. The most common malignancy was melanoma (33.3%) and ovarian cancer took the seventh position (2.6% of all cases). The researchers did not specify information about OC [79]. Similar results were presented by Parazzini et al. The researchers diagnosed 45 ovarian cancer cases among 1475 of all cancers cases in 1,200,263 pregnancies between 2001 and 2012 (eighth position in the ranking of pregnancy-associated cancers), resulting in a risk of pregnancy-associated cancer of 3.7 per 100,000 pregnancies [80]. Smith et al. analyzed 4,846,505 deliveries between 1991 and 1999 in California; they identified 4539 malignancies with 253 cases of ovarian cancer. The highest mortality rate was when the diagnosis of cancer was made between 0 and 3 months before delivery [3]. A clinicopathological analysis of 23 cases of ovarian cancer associated with pregnancy was performed by Behtash and colleagues, in which the patients were treated between 1991 and 2002 at Vali-Asr Hospital. Seventeen pregnant women were diagnosed in Stage I; chemotherapy was administered to 44% of these cases; five patients relapsed and died [27]. In another study, Morikawa et al. analyzed 41 cases of malignant ovarian tumors during pregnancy between 1985 and 2010 in a retrospective study; the researchers focused on pathology-oriented treatment. Thirty-eight pregnant women were diagnosed in Stage I and 12 patients underwent chemotherapy; one of them died due to ovarian cancer [81]. Zhao et al. summarized patients’ experiences of ovarian cancer diagnosed in pregnancy between 1985 and 2003 at Peking Union Medical College Hospital. Twenty-two patients with ovarian malignancy were treated; 16 pregnant women were in Stage I of OC and achieved complete remission, while four of the five patients in an advanced stage died [24]. Only data from these and a few other studies, which are presented in Table 5, satisfactorily described patient outcomes.

### 3.5. Perspectives

Hallum et al. conducted a prospective case–cohort study in which 700 women from the Danish Diet, Cancer and Health cohort participated. The researchers looked for a correlation between microchimerism and ovarian cancer by analyzing women’s blood samples and their responses to questionnaires. In the blood samples, they looked for the presence of a Y chromosome as a marker of male microchimerism. Samples were positive in 65.9% of the control group and in 46% of the ovarian cancer group. The results of this study suggest that male microchimerism reduces the hazard rate of ovarian cancer (HR = 0.44, 95% CI: 0.29–0.68), but the underlying mechanisms are currently unknown [88]. 

A novel idea for OC treatment is viral-based cancer therapy with ofranergene obadenovec (VB-111). VB-111 I is an adenoviral vector which carries the tumor necrosis factor receptor 1-Fas cell surface (TNFR1-FAS), a chimeric death receptor transgene with an altered pre-endothelin 1 promoter, which is activated in angiogenic endothelial cells. It is postulated to restrict tumor angiogenesis through TNF-induced, TNFR1-FAS-mediated endothelial cell apoptosis. The death of the associated tumor cells and antigen release, along with the immune effect of the virus itself may result in enhanced antitumor immunity [89].

Xi et al. reported the high expression of bone morphogenic protein endothelial cell precursor-derived regulator (BMPER) in ovarian cancer, which was detected by immunohistochemistry. This upregulation of BMPER was also related to increased metastasis to lymph nodes and a lower survival rate. It is postulated that the overexpression of BMPER is an independent risk factor of poor prognosis in patients. The suppression of BMPER inhibits the migration, invasion and proliferation of OC, and promotes apoptosis. In that study, the researchers considered that BMPER may become a potential prognostic marker and that modification of this pathway may change the role of BMPER in promoting the malignant biological behavior of ovarian cancer cells [90]. 

Ferreira et al., in their systematic review with meta-analysis, examined the prognostic role of microRNAs (miRNAs) in epithelial OC. They identified 12 miRNAs that may be useful in diagnosis, prognosis and chemotherapy sensitivity in epithelial ovarian cancer management, but further investigations involving prospective randomized trials are needed to validate these data [91].

Lin et al. analyzed the role of the long non-coding RNA (lncRNA) AOC4P in the suppression of metastasis in epithelial ovarian cancer through regulation of the epithelial–mesenchymal transition. It has been found that the expression of AOC4P is decreased in epithelial OC tissues and cell lines. Additionally, this underexpression is positively correlated with lymph node metastasis and the clinico-pathological stage of epithelial ovarian cancer. This anti-metastatic role was also verified in vivo by tumor dissection and bioluminescence imaging. AOC5P may become a novel and probably effective target for anti-metastatic clinical management of epithelial ovarian cancer [92]. The same idea of using lncRNA was presented by Salamini-Montemurri et al. in their review, which analyzed the challenges and opportunities of using lncRNA in clinical practice. In their opinion, lncRNA may create a novel generation of tools in the diagnostics, prognosis and treatment resistance of OC [93]. 

The diversity of ovarian cancer and different risk factors for its development need to be emphasized. OC is related to a specific lifestyle, geographic location or age. Blake et al., in their secondary analysis of a previously prepared systematic review on ovarian cancer diagnosed during pregnancy, noticed the fact that teenage women (≤20 years old) whose pregnancies were complicated by ovarian cancer might be at increased risk of poor survival from OC (adjusted HR = 5.51; 95% CI = 1.29–21.8; *p* = 0.021) [94]. The reason for this observation is still unclear, so new research and analytical insights are needed to solve this problem. 

These different data presented in the articles demonstrate that ovarian cancer concomitant with pregnancy appears to be a very rare occurrence, and only through thorough analysis of all of these cases can we understand the problem and find proper solutions (improved diagnostics and treatment), which can be introduced worldwide. Many logistic and geographical barriers may impede patient access to multidisciplinary tumor boards in referral hospitals, which results in an astounding number of women that get suboptimal care. In order to improve upon this, the “Advisory Board on Cancer in Pregnancy” was created in the Netherlands and France. The two email-based tumor advisory boards consist of highly integrated teams of specialized physicians that remotely discuss clinical cases of cancer in pregnancy and provide advice to other physicians who need expertise on how to manage these female patients [53].

## 4. Conclusions

The basic principles of the management of ovarian cancer during pregnancy are summarized in Table 6. To conclude, pregnancy complicated by ovarian cancer remains a challenge for physicians. Due to a low prevalence of OC in pregnancy and the absence of large randomized trials and major patient cohorts, universal standards of treatment have not yet been proposed. A link between chemotherapy and its influence on the fetus and its later development has been under close examination. The key factor in this situation is the relationship between the doctor’s conscience and the patient’s trust. Decisions concerning the choice of the best management of ovarian cancer are very complex and difficult because of the conflict between the mother’s and the fetus’s wellbeing; thus, a multidisciplinary team consisting of an obstetrician, oncologist, pathologist, anesthesiologist, neonatologist and psychologist is mandatory. 

The aim for pregnant patients is the same as for non-pregnant women regarding malignancy, namely, to survive free of neoplasm for as long as possible. Fortunately, the recurrence-free and overall survival rates for pregnant women are very similar to those reported for non-pregnant patients.

## Figures and Tables

**Figure 1 cancers-12-03795-f001:**
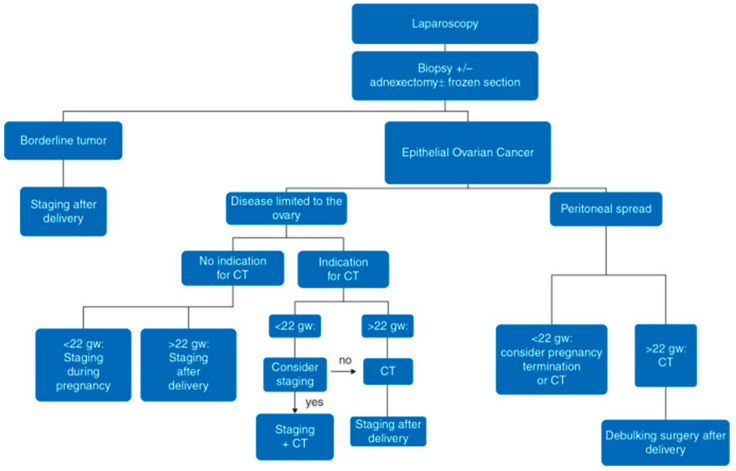
Scheme for the management of epithelial ovarian cancer tumors. According to ESMO guidelines. Staging refers to surgical staging. CT, chemotherapy; gw, gestational weeks [53].

**Figure 2 cancers-12-03795-f002:**
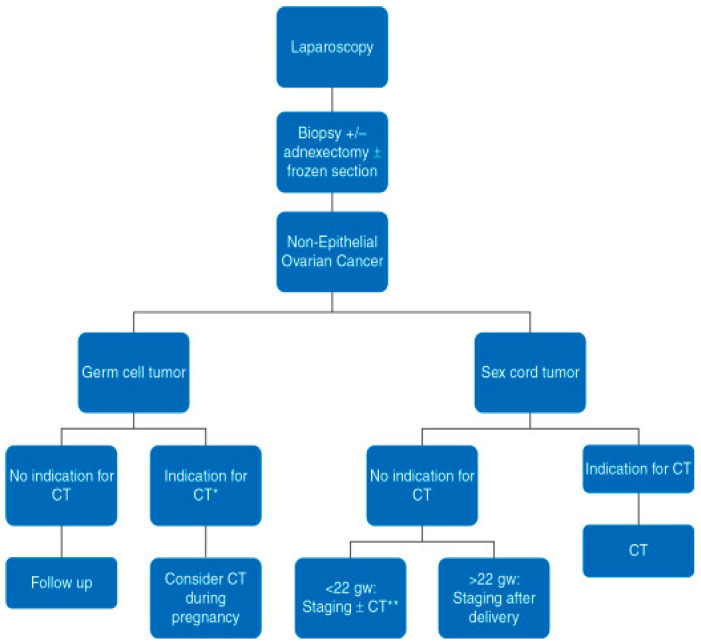
Scheme for the management of nonepithelial ovarian cancer tumors. Staging refers to surgical staging. CT, chemotherapy; gw, gestational weeks. * According to ESMO guidelines; ** CT administered according to restaging surgery findings (Reprinted with permission from Amant, F.; Berveiller, P.; Boere, I.A.; Cardonick, E.; Fruscio, R.; Fumagalli, M.; Halaska, M.J.; Hasenburg, A.; Johansson, A.L.V.; Lambertini, M.; et al. Gynecologic cancers in pregnancy: Guidelines based on a third international consensus meeting. *Ann. Oncol.*
**2019**, *30*, 1601–1612. Copyright Elsevier 2020) [53].

**Table 1 cancers-12-03795-t001:** Frequency of ovarian cancer in pregnancy by tumor histology in the literature.

	Author and Year				
Histopathology	Copeland, 1996 [25]	Zanotti, 2000 [26]	Zhao, 2006 [24]	Behtash, 2008 [27]	Morikawa, 2014 [25]
EOC	37.5%	33–34%	50%	39.1%	81%
Invasive EOC	-	-	22.7%	21.7%	20%
LMP	-	-	27.3%	17.4%	61%
Germ cell tumor	45%	30–33%	40.9%	47.8%	17%
SCT	10%	17–20%	9.1%	13%	2%
Others	7.5%	12–13%	0	0	0

EOC, epithelial ovarian cancer; LMP, low malignant potential tumor; SCT, sex cord stromal tumor.

**Table 2 cancers-12-03795-t002:** Adnexal masses in pregnancy: benign versus malignant features (International Ovarian Tumor Analysis (IOTA) Group Simple Rules) [32].

B-Features	M-Features
B1- Unilocular	M1- Irregular solid tumor
B2- Presence of solid components with a largest diameter of <7 mm	M2- Presence of ascites
B3- Presence of acoustic shadows	M3- At least 4 papillary structures
B4- Smooth multilocular tumor with a largest diameter of <100 mm	M4- Irregular multilocular-solid tumor with a largest diameter of ≥100 mm
B5- No blood flow (color score 1)	M5- Very strong blood flow (color score 4)

**Table 3 cancers-12-03795-t003:** ADNEX MR scoring system [39].

Score 1	No mass visible in MRI
Score 2	Purely cystic ovarian mass Purely endometriotic ovarian mass Purely fatty ovarian mass Ovarian mass without wall enhancement Low b = 1000 sec/mm^2^ –weighted and low T2-weighted signal intensity within solid tissue
Score 3	Mass without solid tissue Curve type 1 within solid tissue
Score 4	Curve type 2 within solid tissue
Score 5	Peritoneal implants Curve type 3 within solid tissue
Score 1 to 3- benign or probably benign; Score 4- indeterminate Score 5- probably malignant	

**Table 4 cancers-12-03795-t004:** Toxicity of the most used chemotherapeutics and their influence on the fetus [65,66].

Drug	Fetal Abnormalities	Comments
Cisplatin	Impaired development, hypoacusia, neutropenia, ventriculomegaly, hair loss	-
Carboplatin	None	-
Paclitaxel	Myelosuppression, pyrolic stenosis	Single cases
Etoposide	Pancytopenia, hypoacusia, secondary leukemias	Particular fear of secondary tumors
Bleomycin	Syndactyly	-
Vinblastine	Syndactyly, plagiocephaly	Safer than etoposide

**Table 5 cancers-12-03795-t005:** Ovarian cancer in pregnancy in the literature and patient outcomes.

Study	Year of Publication	Country	Years	Number of Cases of Ovarian Malignancy	Type of Malignancy (Number)	Stage of Malignancy (Number of Cases)	Relapses (Number)	Deaths (Number)
Cottreau et al. [82]	2019	US, 5 states	2001–2013	44	ovarian cancer	no data	no data	incomplete data
de Haan et al. [73]	2018	Europe, 16 countries	1996–2016	88	ovarian cancer	stage I—66 cases stage II—4 cases stage III—7 cases stage IV—2 cases unknown—9 cases	no data	0 during pregnancy
Parazzini et al. [80]	2017	Italy, Lombardia	2001–2012	45	ovarian cancer	no data	no data	incomplete data
Shim et al. [16]	2016	South Korea	1995–2013	5	ovarian cancer: EOC (4), dysgerminoma (1)	stage I—5 cases	0	1
Zhao et al. [24]	2016	China	1985–2003	22	ovarian cancer: germ cell tumor (9), EOC (5), sex cord stromal tumor (2) + LMP tumor (5)	stage I—16 cases, stage II—1 case, stage III—3 cases, stage IV—2 cases	4	4
Andersson et al. [83]	2015	Sweden	1963–2007	175	ovarian cancer	no data	no data	incomplete data
Nazer et al. [84]	2015	US	2003–2011	180	ovarian cancer (93) + low malignant potential tumor (87)	incomplete data	no data	no data
Morikawa et al. [81]	2014	Japan	1985–2010	41	ovarian cancer: borderline tumor (25), EOC (8), germ cell tumor (7), sex cord stromal tumor (1)	stage I—38 cases, stage II—1 case, stage III—1 case, stage IV—1 case	incomplete data	incomplete data
Eibye et al. [85]	2013	Denmark	1977–2006	74	ovarian cancer	incomplete data	no data	no data
Lee et al. [79]	2012	Australia, New South Wales	1994–2008	47	ovarian cancer	incomplete data	no data	incomplete data
Fauvet et al. [22]	2011	France, 6 centers	1997–2009	40	ovarian cancer: borderline tumor (40)	stage I—35 cases, stage II—2 cases, stage III—2 cases, staging not available—1 case	3	0
Kwon et al. [86]	2010	South Korea	1996–2006	27	ovarian cancer: borderline tumor (15), EOC (7), germ cell tumor (5)	no data	1	0
Van Calsteren et al. [1]	2010	Europe: Belgium, The Netherlands, Czech Republic	1998–2008	4	ovarian cancer	no data	incomplete data	no data
Stensheim et al. [87]	2009	Norway	1967–2002	53	ovarian cancer	incomplete data	no data	11
Behtash et al. [27]	2008	Iran	1991–2002	23	ovarian cancer: EOC (5), germ cell tumor (11), sex cord stromal tumor (3)+ LMP (5)	stage I—17 cases, stage II—1 case, stage III—3 cases, stage IV—1 case	5	5
Leiserowitz et al. [11]	2006	US, California	1991–1999	202	ovarian cancer: EOC (52), germ cell tumor (34), sex cord stromal tumor (1) + low malignant potential tumor (115)	incomplete data	no data	no data
Smith et al. [3]	2003	US, California	1991–1999	253	ovarian cancer	no data	no data	incomplete data
Matsuyama et al. [41]	1989	Japan	1978–1986	6	ovarian cancer: EOC (4), immature teratoma (1), metastatic cancer of colon origin (1)	stage I—4 cases, stage III 1 case, stage IV—1 case	1	1
Haas [68]	1984	GDR	1970–1979	20	ovarian cancer	incomplete data	no data	no data

EOC- epithelial ovarian cancer; GDR- German Democratic Republic; US- United States.

**Table 6 cancers-12-03795-t006:** Ovarian cancer during pregnancy: basic principles of management.

A multidisciplinary team is mandatory. Pregnant women should be informed, in great detail, about the benefits and risks of treatment. Patients’ beliefs and wishes should be acknowledged. The management of pregnant patients should take into consideration the physiological changes during pregnancy. Numerous imaging procedures using radiography are not harmful to the fetus when they are used with proper shielding. Laparoscopic or open surgery can be safe, providing they are executed by “proper hands”. Systemic chemotherapy should be started from the second trimester if possible. Most chemotherapeutics are safe during the second and third trimesters. There is no evidence based medicine about targeted therapies. The doses of medicaments should be the same for pregnant women as for non-pregnant women. Radiotherapy is contraindicated during pregnancy. Termination of pregnancy may be considered in the case of immediate treatment. No change in prognosis has been shown after the termination of pregnancy. Differences in the rates of survival may exist between pregnant and non-pregnant women with cancer. There is no evidence that subsequent pregnancy increases the risk of recurrence of the disease.

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
