# Peer review of "Ovarian Cancer and Pregnancy—A Current Problem in Perinatal Medicine: A Comprehensive Review"

_cancers, 2020, doi:10.3390/cancers12123795_

Round 1
Reviewer 1 Report
Dear editor,
Thank you for the opportunity to revise this manuscript.
In my opinion, as you will read in the comments, minor revisions are needed.
For any further question don’t hesitate to contact me.
The authors revise and discuss literature data regarding ovarian cancer during pregnancy also presenting important recommendations for the diagnosis and treatment.
In my opinion the manuscript is worth for publication however, the following minor revisions are needed before acceptance:
- Extensive English revisions: several minor spelling error throughout the manuscript must be corrected (i.e. most sentence starting with “the” such as the ovarian cancerare incorrect). I suggest language revisions by a native speaker so that data presented become more comprehensible.
- Introduction and discussion section can be shortened focusing only on data regarding pregnancy-related ovarian tumors. Many presented data such as epidemiological risk factors and genetics are not so relevant to the aim of the study.
- In the introduction section all ovarian cancer histotypes are well discussed “The EOC is the vast majority of ovarian cancers in non-pregnant women, but it is responsible for 54 only 35% cases of OC during pregnancy [20]. Much more common are borderline, germ cell and sex 55 cord tumors, which is involved in their increased prevalence in reproductive age [21]. Borderline 56 tumors are neoplasms of low malignant potential, but in pregnant women they have higher incidence 57 of more aggressive histologic features like microinvasion [22]. The two most common germ cell 58 tumors in pregnancy are dysgerminoma and yolk-sack tumor [21]. Five subtypes of epithelial ovarian 59 cancers can be distinguished: high- grade serous, low-grade serous, endometrioid, clear cell and 60 mucinous ovarian cancer”. However, to improve the manscript quality, an additional table, reporting the frequency of all discussed histotypes would be useful for readers.
Author Response
Reviewer 1
We would like to thank the Members of the Editorial Team for their assistance during submitting our article but mainly to the Reviewers for their excellent work.
Please let us refer to the comments of Reviewer 1.
1. Extensive English revisions: several minor spelling error throughout the manuscript must be corrected (i.e. most sentence starting with “the” such as the ovarian cancer are incorrect). I suggest language revisions by a native speaker so that data presented become more comprehensible.
Response: We appreciate the Reviewer’s comment. The manuscript was revised by native speaker. We hope that data are presented more comprehensible now.
2. Introduction and discussion section can be shortened focusing only on data regarding pregnancy-related ovarian tumors. Many presented data such as epidemiological risk factors and genetics are not so relevant to the aim of the study.
Response: We appreciate the reviewer’s comment. We do believe that the data mentioned above by the Reviewer let us present the problem in larger context so in our opinion it is worth including it into the text. According to the Rewiewer’s suggestion we shortened the Introduction. But, we have to remember about the limit of words: at least 4000 of main text. If we shorten introduction or discussion we won’t meet the criteria of manuscript submission.
3. In the introduction section all ovarian cancer histotypes are well discussed “The EOC is the vast majority of ovarian cancers in non-pregnant women, but it is responsible for 54 only 35% cases of OC during pregnancy [20]. Much more common are borderline, germ cell and sex 55 cord tumors, which is involved in their increased prevalence in reproductive age [21]. Borderline 56 tumors are neoplasms of low malignant potential, but in pregnant women they have higher incidence 57 of more aggressive histologic features like microinvasion [22]. The two most common germ cell 58 tumors in pregnancy are dysgerminoma and yolk-sack tumor [21]. Five subtypes of epithelial ovarian 59 cancers can be distinguished: high- grade serous, low-grade serous, endometrioid, clear cell and 60 mucinous ovarian cancer”. However, to improve the manscript quality, an additional table, reporting the frequency of all discussed histotypes would be useful for readers.
Response: Thank you for this suggestion and the appropriate table has been prepared.
Reviewer 2 Report
Good review, fairly well written, even if language should be improved by a native english speaker revision.
I would suggest to add a paragraph describing the oncological outcome of patients with ovarian cancer in pregnancy; probably a table summarizing the number of patients considered in each paper, the type of malignancy, the stage and the reported events (relapse, death) would be very useful to the reader.
Author Response
Reviewer 2
We would like to thank the Members of the Editorial Team for their assistance during submitting our article but mainly to the Reviewers for their excellent work.
Please let us refer to the comments of Reviewer 2.
1.Good review, fairly well written, even if language should be improved by a native English speaker revision.
Response: We appreciate the Reviewer’s comment. The manuscript was revised by native speaker. We hope that data are presented more comprehensible now.
2. I would suggest to add a paragraph describing the oncological outcome of patients with ovarian cancer in pregnancy; probably a table summarizing the number of patients considered in each paper, the type of malignancy, the stage and the reported events (relapse, death) would be very useful to the reader.
Response: We appreciate the Reviewer’s comment. The paragraph on the outcome of patients and the table has been prepared and enclosed in our manuscript. However, only some of the papers contained data that would allow them to be presented in a table, so the number of these articles is very limited.
Reviewer 3 Report
Re: The manuscript entitled “Ovarian cancer and pregnancy - current problem in perinatal medicine: a comprehensive review.” By Dominik Franciszek Dłuski et al.
This review presents information and discusses the problem of ovarian cancer complicating pregnancy and the current recommendations for the diagnostics and treatment with the emphasis on their risk for the fetus. Due to the rareness of the diseases and limited literature covered in this review, this review may be more interesting to more specialized researchers and clinicians.
There are several issues that need to be revised/addressed.
- The section “Summary and conclusions” is not written as it is indicated. It needs to be re-organized, with part of it moved to the main section.
- Part of Introduction directly repeats what is in Abstract, which needs to be revised.
- P2We analyzed all types of articles (case reports, clinical trials, reviews, systematic reviews, meta-analyses) because of the rarity of discussed problems.
- Treatment of pregnant women with chemotherapy is considered harmless. This conclusion is only based on a few studies, which should be carefully considered and more research should be encouraged.
- Page 9 “Next the duplicates were removed (551 articles), 414 articles were excluded based on the titles and abstracts, leaving 137. After reading the full text articles 81 paper were excluded and 56 articles had useful information, which we could use to prepare this article. The results are presented in four sections: diagnostics, management, obstetric outcome and perspectives.” The criteria for these selections need to be clearly stated, which is very important.
- The English needs to be edited throughout the manuscript. Only a few examples are mentioned here.
- In many places, a semicolon needs to be added.
- “In the other hand” should be “on the other hand”
- Page 8 “Very important is the …” should be revised.
- Page 9 “37 centers from 16 countries taking part in the International Network on Cancer, …”. A verb is needed for the sentence.
Author Response
Reviewer 3
We would like to thank the Members of the Editorial Team for their assistance during submitting our article but mainly to the Reviewers for their excellent work.
Please let us refer to the comments of Reviewer 3.
1. The section “Summary and conclusions” is not written as it is indicated. It needs to be re-organized, with part of it moved to the main section.
Response : According to the Reviewer’s opinion the part of this chapter was moved to the main section. The title of this section was changed to “Conclusions”.
2. Part of Introduction directly repeats what is in Abstract, which needs to be revised.
Response : We appreciate the reviewer’s comment. According to the Reviewer’s opinion, Abstract has been revised and the part below:
“The delay in planning pregnancy to later reproductive age has been observed for the last decades. The incidence of many malignancies increases with age since the number of pregnant women with the oncologic disease is expected to rise as well: this challenging problem complicates 1 in 1000 pregnancies and is becoming more common currently” has been removed.
3. P2 We analyzed all types of articles (case reports, clinical trials, reviews, systematic reviews, meta-analyses) because of the rarity of discussed problems.
Response: We appreciate the Reviewer’s comment and have revised the original sentence (at line 89-91): to below “We analyzed types of articles as case reports, clinical trials, population-based studies, reviews, systematic reviews and meta-analyses, due to the rarity character of the discussed problem.”
4. Treatment of pregnant women with chemotherapy is considered harmless. This conclusion is only based on a few studies, which should be carefully considered and more research should be encouraged.
Response: We appreciate the Reviewer’s comment. The chemotherapy during pregnancy seems to be relatively harmless but this issue is still to be elucidated.
5. Page 9 “Next the duplicates were removed (551 articles), 414 articles were excluded based on the titles and abstracts, leaving 137. After reading the full text articles 81 paper were excluded and 56 articles had useful information, which we could use to prepare this article. The results are presented in four sections: diagnostics, management, obstetric outcome and perspectives.” The criteria for these selections need to be clearly stated, which is very important.
Response: We appreciate the Reviewer’s comment and have revised the original sentences (at line 94-105) to below:
“A computerized literature search in electronic databases PubMed/MEDLINE was performed to identify articles on ovarian cancer and pregnancy using the keywords: ,,ovarian cancer", ,,ovarian malignancy", ,,adnexal masses", ,,ovarian tumor", and ,,pregnancy". Only publications in English were considered. Since the discussed problem is extremely rare the different types of articles as case reports, clinical trials, reviews, systematic reviews, and meta-analyses were analyzed. We initially identified a total of 1319 papers. Next the duplicates were removed and 551 articles was left for our assessment, 414 articles were excluded based on the titles and abstracts, leaving 137. After reading the full text articles, 81 papers were excluded from analysis because of incomplete case reporting, or the inability to assign individual diagnostic, clinicopathological data, treatment regimens, or outcome. We also excluded articles only devoted to technical aspects of surgical management and detailed histopatological analysis. The results of final review of 56 articles are presented in five sections: diagnostics, management, obstetric outcome, patients’ outcome and perspectives.”
6. The English needs to be edited throughout the manuscript. Only a few examples are mentioned here.
Response: We appreciate the reviewer’s comment. The manuscript was revised by native speaker. We hope that data are presented more comprehensible now.
7. In many places, a semicolon needs to be added.
Response: We appreciate the Reviewer’s comment. While revising the manuscript the punctuation marks have been added if necessary.
8. “In the other hand” should be “on the other hand”
Response: We appreciate the Reviewer’s comment and have revised the original part of sentence (at line 112-113) to below:
“On the other hand..”
9. Page 8 “Very important is the …” should be revised.
Response: We appreciate the Reviewer’s comment. The manuscript was revised by native speaker. We hope that data are presented more comprehensible now.
We changed this sentence (at line 287-289) to below: “The assessment of the peak systolic velocity (PSV) in MCA is of great clinical value in the care of high-risk pregnancies in the diagnosis and management of fetal anemia and intrauterine growth restriction (IUGR)”.
10. Page 9 “37 centers from 16 countries taking part in the International Network on Cancer, …”. A verb is needed for the sentence.
Response: We appreciate the reviewer’s comment and have revised the original sentence (at line 301-303) to below:
“1170 pregnant women with cancer were reported by de Haan et al. in descriptive cohort study with results from over the 20 years of research; 37 centers from 16 countries participated in the International Network on Cancer, Infertility and Pregnancy (INCIP).”
Round 2
Reviewer 3 Report
The authors have responded to my concerns. However, the English is still need to be improved.
